# Correlation between Lindane Use and the Incidence of Thyroid Cancer in the United States: An Ecological Study

**DOI:** 10.3390/ijerph192013158

**Published:** 2022-10-13

**Authors:** Mathilda Alsen, Vikram Vasan, Eric M. Genden, Catherine Sinclair, Maaike van Gerwen

**Affiliations:** 1Department of Otolaryngology-Head and Neck Surgery, Icahn School of Medicine at Mount Sinai, New York, NY 10029, USA; 2Institute for Translational Epidemiology, Icahn School of Medicine at Mount Sinai, New York, NY 10029, USA

**Keywords:** endocrine disruptive chemicals, EDCs, thyroid cancer, pesticides, lindane, environmental health, epidemiology

## Abstract

The increasing rate of thyroid cancer may be attributable to endocrine disruptive chemicals. Lindane is a persistent organochlorine pesticide with endocrine disruptive properties that has been classified as carcinogenic to humans. The aim of this ecological study was to evaluate potential correlation between lindane exposure and thyroid cancer incidence in the United States (US). Data on statewide age-adjusted thyroid cancer incidence rate (per 100,000 people) was obtained from the Centers for Disease Control and Prevention for all US states for 2019. Lindane use estimates per cropland (kg/acres cropland) were then overlaid on the map of age-adjusted thyroid cancer incidence rate using ArcGIS. The trend of lindane use in the US between 1992 and 2007 was calculated using the Mann Kendall correlation test. The correlation between statewide lindane use and age-adjusted thyroid cancer incidence rates was calculated using Spearman correlation. Lindane use in the US decreased significantly between 1992 and 2007 (T = −0.617; *p* < 0.001). There was no statistically significant correlation between lindane use in 1992 and thyroid cancer incidence rate for any of the years between 1999 and 2019. Our results suggest that restrictions clearly seem to be effective in reducing lindane use, however, more research is needed for individual pesticides and thyroid cancer.

## 1. Introduction

The incidence rate of thyroid cancer has increased significantly in recent decades in the United States (US) [1]. The increased incidence may be partially explained by the early detection of small, mainly papillary thyroid cancers (PTC) due to more frequent use of advanced imaging technologies, so- called overdiagnosis [2]. Although overdiagnosis- by definition- leads to the detection of small PTCs in early stage and should thus result in a decline of thyroid cancer specific mortality, studies have reported that the overall thyroid cancer mortality remained stable and mortality of distant or advanced stage thyroid cancer even increased [2]. Overdiagnosis has been estimated to explain ~50% of the increase in PTC incidence, so research into other risk factors that would explain this increase is needed [2].

Although previous epidemiological studies suggested that approximately 40% of thyroid cancer diagnoses in the US may be attributable to environmental factors, research investigating the role of environmental factors in cancer risk is limited [2]. Chemicals with endocrine disruptive properties are potential risk factors, because of their ability to mimic the thyroid hormones in the body [2,3]. Organochlorine pesticides (OCPs) are known endocrine disrupting chemicals. OCPs were widely used for residential and agricultural purposes since the 1940s until some of them, including dichlorodiphenyltrichloroethane (DDT) and hexachlorobenzene (HCB), were banned or restricted in the 1960s and 1970s because of their toxicity and carcinogenic effects [4,5,6]. OCPs are known to alter thyroid hormone levels due to their similar structure to triiodothyronine (T3) and thyroxine (T4) [6,7]. Existing literature has shown that exposure to specific OCPs has been associated with various types of malignancies [8,9], as was exposure to chlordane exposure associated with rectal cancer, dieldrin exposure with lung cancer, toxaphene exposure with melanoma, and chlordane/heptachlor exposure with leukemia [8]. One study found endosulfan and 4,4,-DDE exposure to be significantly associated with breast and blood cancer [10]. Another study has reported a positive association between heptachlor, methoxychlor, DDT, and HCB exposure, and breast cancer [11]. Several epidemiological studies investigating the association between OCPs and thyroid cancer risk showed conflicting results, although some literature supports the hypothesis that pesticides could play a major role in the thyroid cancer etiology [3,12].

In OCP, lindane is the y-isomer of the hexachlorocyclohexane (HCH) [13]. Lindane was extensively used as an insecticide for wood, crops, livestock, and seed, starting in the 1950s; however lindane use was restricted in agriculture between 1970 and 1990 [13], and discontinued for agricultural purposes in the US in 2006 [14]. Lindane is, however, still being formulated and used in prescription lotions, cream, and shampoos for head lice and scabies [14]. The most common acute side effects of lindane include seizures, dizziness, and headache, although high doses may be fatal [15]. The International Agency for Research on Cancer (IARC) has classified lindane as carcinogenic to humans (Group 1), while the EPA has classified lindane as a possible human carcinogen (B2/C) [14,16]. Lindane is listed in the Stockholm convention under Annex A (elimination), concluding that lindane is likely to lead to significant adverse human health and environmental effects [17]. Exposure to lindane is significantly associated with non-Hodgkin’s lymphoma [18]. Blair et al. reported a 50% increased risk of non-Hodgkin’s lymphoma among agricultural workers exposed to lindane [19].

Lindane has been detected in air, soil, water, and sediment, and can be absorbed into the body through inhalation and dermal exposure, but mostly through diet [20]. It is a persistent type of OCP and concentrates in the environment mostly in water, but can also be found in soil and air [21]. The reported half-life of lindane in the environment is long and varies between 11 years in seawater and 100 years in the Arctic Ocean [22]. The half-life of lindane in soil is approximately 2 weeks [20]. The isomers of lindane are lipohilic, allowing accumulation in fatty tissues for long periods; the beta isomer (beta-HCH) has a half-life of about seven years in fatty tissues [20].

To the best of our knowledge, only one study investigated the association between lindane exposure and thyroid cancer risk, and found a significantly positive association between lindane exposure and thyroid cancer risk among male pesticide users (hazard ratio (HR) = 1.74, confidence interval (CI): 1.06–2.84) [9]. Understanding the potential association between OCP exposure and thyroid cancer risk is imperative as some OCPs are still being used to date [23]. Given that lindane is a carcinogen and exposure to lindane has been associated with thyroid dysfunction, this study assessed the potential correlation between lindane exposure and thyroid cancer in the US by performing an ecological study of thyroid cancer incidence and lindane use.

## 2. Materials and Methods

Data on statewide age-adjusted thyroid cancer incidence rate (per 100,000 people) was obtained by gender from the Centers for Disease Control and Prevention (CDC) for all US states for 2019, the most recent year with this data available. To examine any possible link between lindane use and thyroid cancer before year 2019 we looked at the correlation between lindane use (kg/acres of cropland) in 1992 and statewide thyroid cancer incidence rates for each year between 1999 and 2019 for males and females combined. This incidence data has been compiled from selected cancer registries meeting US Cancer Statistics data quality criteria covering 99% of the US population [24]. The data includes all ages, races/ethnicities, and sex. The data was imported into the ArcGIS, a mapping software developed by the Environmental Systems Research Institute (version 10.8; ESRI, Redlands, CA, USA) [25].

County-level lindane use was obtained from the United States Geological Survey (USGS). Lindane use was estimated using two estimation methods in the USGS survey: EPest-low and EPest-high. These estimation methods, developed by Thelin and Stone (2013) [26], calculate the estimated pesticide use (EPest) rate use for each crop acre by year using surveys from the Crop Reporting Districts (CRD) in conjunction with data reported by the US Department of Agriculture National Agricultural Statistics Service (NASS) for all states except California [26]. EPest-high includes more counties with estimated data compared to EPest-low: EPest-high estimates the use based on pesticide-by-crop use in surrounding areas whereas EPest-low assumes zero use for missing data. EP-pest high estimates were used in the present study as this represents a better estimate of potential lindane exposure of the population.

To the best of our knowledge, the latency period between lindane/pesticide exposure and thyroid cancer is unknown. To take latency into consideration, the earliest year available for the lindane data (1992) was included in the present study. Furthermore, lindane data for the year 2007 was explored because this was the first year following the discontinuation of lindane use in 2006. For the included years, annual lindane use estimates were reported as kilograms (kg) by county, which was then summarized by state and divided by total acres cropland/1000 (kg/acres cropland). Data on the cropland acreage per state was obtained from the US Department of Agriculture (USDA) for the included years.

### Statistical Analysis

Trends of overall lindane use between the years 1992 and 2007 was calculated using the Mann Kendall correlation test. Lindane use, presented in quintiles for the years 1992 and 2007, was then overlaid on the map of age-adjusted thyroid cancer incidence rate (2019) using the ArcGIS. The correlation between statewide lindane use (kg/acres cropland) for the years 1992 and 2007 and age-adjusted thyroid cancer incidence rates for the year 2019 was calculated using Spearman correlation by gender. The correlation between lindane use (kg/acres of cropland) in 1992 and statewide thyroid cancer incidence rates for each year between 1999 and 2019 was also calculated for males and females combined. Analyses were performed using the R software (version 4.1.2; Vienna, Austria).

## 3. Results

The age-adjusted thyroid cancer incidence rates in 2019 ranged from 8.0 to 19.6 cases per 100,000 persons, with the lowest age-adjusted thyroid cancer incidence rate in Mississippi and the highest reported thyroid cancer incidence rate in New York. For males, the age-adjusted incidence rate ranged from 4.50 to 10.8 cases per 100,000 persons with the lowest age-adjusted thyroid cancer incidence rate in Mississippi and the highest age-adjusted reported thyroid cancer incidence rate in New York. The age-adjusted incidence rates ranged from 11.40 to 30.1 cases per 100,000 persons for females with the lowest age-adjusted thyroid cancer incidence rate in Mississippi and the highest reported age-adjusted thyroid cancer incidence rate in North Dakota. Data on thyroid cancer incidence rates were not available for the state of Nevada. (Figure 1, Figure 2 and Figure 3) (Appendix A).

There was no significant correlation between lindane use in 1992 and thyroid cancer incidence rates between 1999–2019 for males and females combined (Table 1). Overall lindane use decreased significantly between 1992 and 2007 (T = −0.617; *p* < 0.001) (Table 2). In 1992, the lindane estimates ranged from 0.0003 to 2.33 kg/ cropland in Washington and Georgia, respectively. In 2007, the lindane estimates ranged from 0.0015–0.80 kg/cropland in Oklahoma and Delaware, respectively (Appendix A). There was no significant correlation between lindane use in 1992 and thyroid cancer incidence rate in 2019 for males and females combined (r = 0.17; *p* = 0.364), for males (r = 0.17; *p* = 0.368) or females (r = 0.17; *p* = 0.368) (Figure 4). There was no significant correlation between lindane use in 2007 and thyroid cancer incidence rates in 2019 for males and females combined (r = −0.17; *p* = 0.327), for males (r = −0.17.; *p* = 0.329) or females (r = −0.203; *p* = 0.234) (Appendix A).

## 4. Discussion

This first ecological study investigating a potential correlation between lindane use and thyroid cancer incidence showed clear statewide differences in the patterns of lindane use as well as thyroid cancer incidence rates. The overall lindane use decreased significantly between 1992 and 2007, likely indicating the effect of implemented restrictions in lindane use. While there was no evident correlation between thyroid cancer and lindane use, it is worth noting that certain states had high lindane use and high thyroid cancer rates in the northeast area, including Delaware, Rhode Island, New York, and New Jersey.

The present study highlights the effectiveness of pesticide restrictions with lindane use trending downwards over time. The agricultural use of lindane has been restricted nationwide since the 1970s, but California was the first state to ban the pharmaceutical use of lindane in 2002, after detecting lindane in wastewater, which then entered lakes, rivers, and the ocean. California engineers calculated that one single lice or scabies treatment containing lindane would contaminate 6 million gallons of water [27]. After the ban in California in 2002, other states including New York, Washington, Maine, and Michigan, began similar discussions about a potential lindane ban [28]. In 2005, the WHO confirmed that the carcinogenicity of lindane [29], and in 2006 Lindane was banned in the US for agricultural purposes by the EPA [14]. Nevertheless, the use of lindane as a lice or scabies treatment is still approved by the US Food and Drug Administration (FDA) [14]. In the EU, all uses of lindane have been banned [22]. Overall, lindane is currently banned in 52 countries and restricted in 33 countries [29].

There are some environmental factors that have been shown to be directly linked to thyroid cancer risk [4]. Exposure to radiation has been proven carcinogenic, where the thyroid is particularly susceptible to radioiodine exposure [4]. Radioactive exposure can lead to DNA lesion and cell death [4]. One study reported that thyroid cancer has been more prevalent in children ages 0–4 years who lived in an area with high levels of ionizing radiations [30]. Other environmental exposures such as flame-retardants, phthalates, polyfluoroalkyl substances (PFAS) and polychlorinated biphenyls (PCBs) are known EDCs with thyroid-disruptuve properties, and interfere with endocrine signalling at cellular and molecular levels [3]. Metals such as cadmium, manganese and lead induce inflammation and the immune response to autoantigens. They also induce inflammatory reactions in the thyroid [4]. Air pollution has shown to have a similar inflammatory response to autoantigens and produce reactive oxygen species [4].

Lindane may exert thyroid-related carcinogenic effects through the generation of oxidative stress, [31], which has shown to be an important contributor to the formation and progression of cancer [32], and through increased mutations and DNA damage [33]. According to IARC, there is weak evidence that lindane alters cell proliferation or death, or induces chronic inflammation [13]. Furthermore, there is moderate evidence that lindane modulates receptor-mediated effects and that it is genotoxic [14]. Exposure to lindane treatment in human cells in vitro showed a chromosomal change, sister-chromatid exchange, and increased micronucleus formation, while results were mixed or negative for DNA-adduct formation and other types of DNA damage. There is no reliable data on genotoxicity in humans [14]. Nevertheless, there is strong evidence that lindane is immunosuppressive and that it induces oxidative stress [13]. An animal study reported excess thyroid C-cell adenoma among female rats after being exposed to lindane [34]. Lindane use in humans and the association with thyroid cancer was investigated using the Agricultural Health Study (AHS), a prospective cohort including 57,310 licensed private and commercial pesticide applicators. The study reported a positive association between lindane exposure and thyroid cancer among male participants (hazard ratio (HR) = 1.74, confidence interval (CI): 1.06–2.84) [9]. It is therefore of importance to further evaluate the effects of lindane exposure among those who live, or work, in high risk areas, such as farmers and their families, since occupational exposure poses a unique risk.

Although chronic, low dose exposure may be more difficult to measure and evaluate, some evidence suggests that chronic exposure to low dose lindane may be of greater health concern resulting in neurobehavioral, neurochemical, and electrophysiological effects, due to induced alteration in GABA receptor activity [35]. Multiple studies have investigated the effects of lindane on the thyroid gland. Prenatal exposure to beta-HCH, a transformable isomer of lindane, has been shown to alter thyroid hormones in newborns [36]. An increased risk of hypothyroidism was found among female spouses of pesticide applicators (adjusted odds ratio (ORadj) = 1.2 (95% CI: 1.0, 1.6)) after exposure to several OCPs combined, where an elevated but non-significant increased odds was found for hypothyroidism after being exposed to lindane (ORadj = 1.5 (95% CI: 0.93, 2.4)), further highlighting the importance of examining highly exposed populations including their family members [37]. Furthermore, a significant positive association between lindane exposure and hypothyroidism was found among pesticide applicators of 62 years and older (HR = 1.54; 95% CI: 1.23, 19.4) [38]. Thyroid dysfunction following lindane exposure has been confirmed in animal studies reporting decreased thyroid hormones after exposure to lindane [14,18,39].

Exposure to lindane and other OCPs during critical developmental periods in early life may negatively impact thyroid health, which has been assessed in some human and animal studies. A study reported that exposure to DDT metabolites before the age of 14 is associated with breast cancer in young women, confirming the need to evaluate the role of OCPs exposure during developmental periods [40]. Evidence has been provided that lindane transfers through the placenta. Prenatal lindane exposure in the offspring of exposed rats induced postnatal alterations in xenobiotic-metabolizing cytochrome P450s in the brain and liver [41]. Other OCPs also pass the placenta as a positive association was found between exposure to DDT and HCH during pregnancy and intra-uterine growth retardation [32,41,42]. Animal studies have found an increased risk of altered neurochemical and behavioral effects in developing rats after prenatal exposure to lindane, as well as long-lasting effects on reproduction later in life [43,44]. However, little is known about lindane exposure during critical developmental windows in utero and early life, and the risk of thyroid disease/cancer later in life, thus warranting further investigation.

To the best of our knowledge, this is the first ecological study investigating the association between lindane exposure and thyroid cancer risk in large representative populations. Therefore, we believe our report provides valuable new information in relation to the effects of lindane use. Strengths include the large number of states included in the study, despite some missing data for a few states. This study has several limitations. First, ecological studies are observational and look at entire populations in question, so no conclusions can be drawn at individual level nor on causality. Ecological studies are also not useful for hypothesis testing because of potential uncontrolled confounding. It is important to note that lindane application does not necessarily result in actual exposure at individual level. In addition, CDC only include all thyroid cancer histology types and no distinction of different types of thyroid cancer is made. It would be beneficial for future studies to examine lindane exposure and its association with different aggressive histological variants.

## 5. Conclusions

Although no correlation between statewide lindane use and thyroid cancer risk was found, and lindane use trended downwards following restrictions and bans, future studies are needed to explore the effects of lindane exposure on the thyroid gland, mainly given the carcinogenic and endocrine disruptive properties of lindane combined with the long environmental half-life of lindane leaving the population at risk of exposure long after usage restrictions. Research is particularly warranted in areas of high exposure and among highly exposed populations, including long-term, low-dose exposure to lindane, as well as in combination with exposure to other OCPs.

## Figures and Tables

**Figure 1 ijerph-19-13158-f001:**
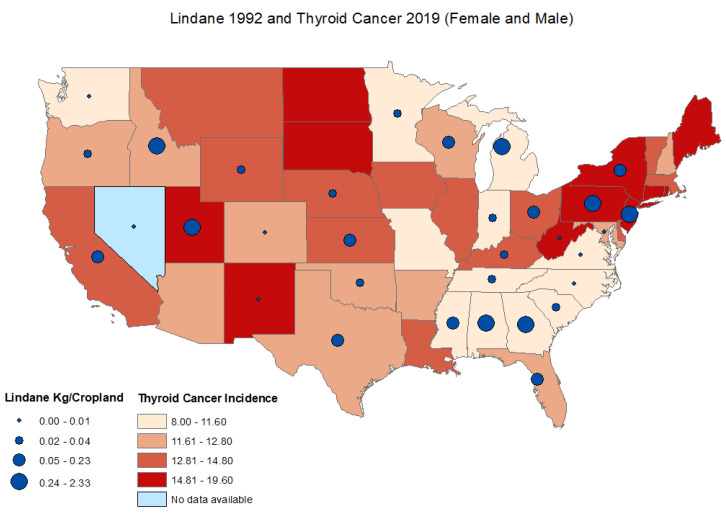
Distributions of age-adjusted thyroid cancer incidence rates per state and estimated lindane high-use (kg/Cropland) male and female.

**Figure 2 ijerph-19-13158-f002:**
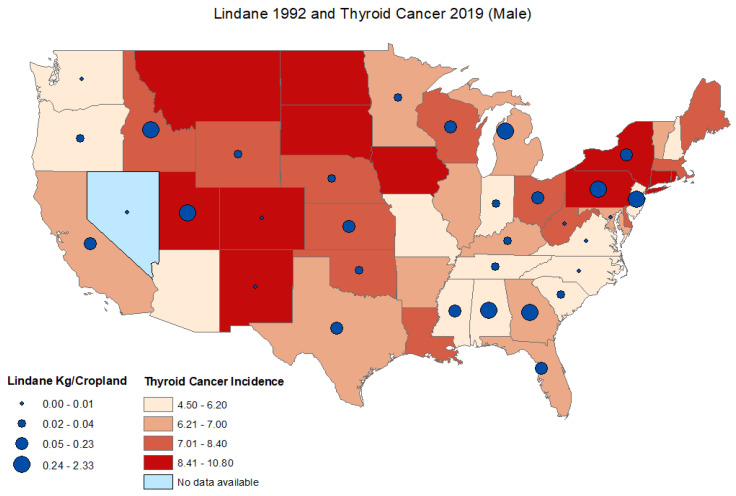
Distribution of age-adjusted thyroid cancer incidence rates per state and estimated lindane high-use (kg/Cropland) male.

**Figure 3 ijerph-19-13158-f003:**
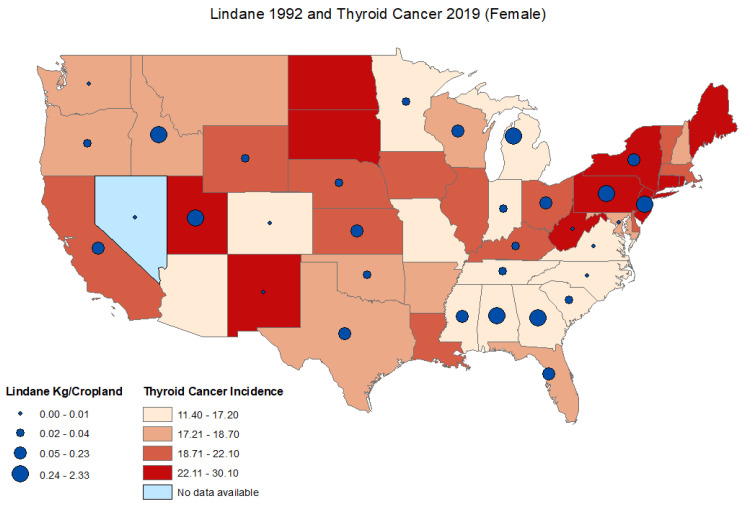
Distribution of age-adjusted thyroid cancer incidence rates per state and estimated lindane high-use (kg/Cropland) female.

**Figure 4 ijerph-19-13158-f004:**
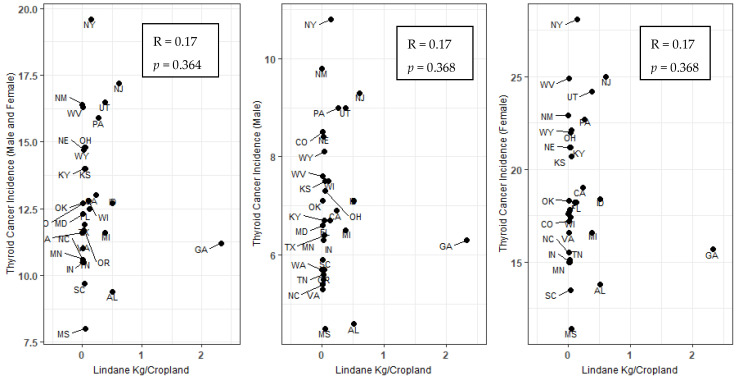
Spearman correlation of age-adjusted thyroid cancer incidence rates per state (2019) and statewide lindane use (kg/Cropland) (1992).

**Table 1 ijerph-19-13158-t001:** Lindane use in 1992 correlated with thyroid cancer incidence rates 1999–2019.

Year	Correlation Coefficient	*p*-Value
1999	0.021	0.911
2000	0.109	0.567
2001	0.059	0.753
2002	0.122	0.521
2003	0.133	0.482
2004	0.110	0.561
2005	0.06	0.764
2006	0.02	0.929
2007	0.05	0.791
2008	0.01	0.954
2009	0.06	0.756
2010	0.07	0.716
2011	0.07	0.725
2012	0.06	0.742
2013	−0.04	0.838
2014	0.13	0.482
2015	0.09	0.623
2016	0.14	0.453
2017	0.04	0.821
2018	0.09	0.611
2019	0.17	0.364

**Table 2 ijerph-19-13158-t002:** Overall lindane use between 1992 and 2007.

Year	Mann-Kendall	*p*-Value
1992–2007	−0.617	<0.001

## Data Availability

Publicly available datasets were analyzed in this study. This data can be found here: Data|U.S. Geological Survey at https://www.usgs.gov/products/data and USCS Data Visualizations—CDC at https://gis.cdc.gov/Cancer/USCS/#/AtAGlance/ (accessed on 1 September 2022).

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
