# Peer review of "Correlation between Lindane Use and the Incidence of Thyroid Cancer in the United States: An Ecological Study"

_ijerph, 2022, doi:10.3390/ijerph192013158_

Round 1
Reviewer 1 Report
I read with great interest the manuscript entitled "Correlation between lindane use and the incidence of thyroid cancer in the United States; an ecological study". The manuscript is well written and it is a very current topic.
Please discuss further the environmental among other factors which are implied in the pathogenesis of thyroid cancer.
Please review either the introduction or the discussion section referring to the following manuscript "Increased trend of thyroid cancer in childhood over the last 30 years in EU countries: a call for the pediatric surgeon, C Spinelli, M Ghionzoli, C Oreglio, B Sanna… - European Journal of Pediatrics, 2022". The mentioned paper is elucidating the incidence of thyroid cancer within the pediatric population in Europe.
Author Response
Reviewer #1: I read with great interest the manuscript entitled "Correlation between lindane use and the incidence of thyroid cancer in the United States; an ecological study". The manuscript is well written and it is a very current topic.
Thank you very much for providing a comprehensive review of our manuscript and recommendations to improve the study. Below, you will find our point-by-point answers to the comments raised by the reviewer. The provided page numbering refers to the clean version of the revised manuscript.
- Please discuss further the environmental among other factors which are implied in the pathogenesis of thyroid cancer. Please review either the introduction or the discussion section referring to the following manuscript "Increased trend of thyroid cancer in childhood over the last 30 years in EU countries: a call for the pediatric surgeon, C Spinelli, M Ghionzoli, C Oreglio, B Sanna… - European Journal of Pediatrics, 2022". The mentioned paper is elucidating the incidence of thyroid cancer within the pediatric population in Europe.
Thank you for this comment. We have added information that discusses environmental factors implied in the pathogenesis of thyroid cancer including other environmental chemicals as well as air pollution. We added a short paragraph on radioactive exposures which is one of the few known environmental risk factors of thyroid cancer risk. We have added this paragraph to the discussion section (page 8).
“There are some environmental factors that have shown to be directly linked to thyroid cancer risk. Exposure to radiation has been proven carcinogenic, where the thyroid is particularly susceptible to radioiodine exposure. (4) Radioactive exposure can lead to DNA lesion and cell death. One study reported that thyroid cancer has been more prevalent in children ages 0-4 years who lived in an area with high levels of ionizing radiations. (30) Other environmental exposures such as flame-retardants, phthalates, Polyfluoroalkyl substances (PFAS) and Polychlorinated Biphenyls (PCBs) are known EDCs with thyroid-disruptuve properties, and interfere with endocrine signalling at cellular and molecular levels. (3) Metals such as cadmium, manganese and lead induce inflammation and the immune response to autoantigens. They also induce inflammatory reactions in the thyroid. (4) Air pollution has shown to have a similar inflammatory response to autoantigens and produce reactive oxygen species. (4)”
Reviewer 2 Report
Dear authors,
Your study is fairly conducted and the results are well presented. Still, I have some concerns regarding the fact that you did not evaluate the type of thyroid cancer, given the fact these types have many differences in growing behavior. Maybe you can address this in a paragraph, searching if there are certain types of thyroid cancer that are more prone to develop. Also, I think you have to write a paragraph about the strengths of this study; you have already stated the limitations, so a section on strengths should be welcomed.
Good luck!
Author Response
Reviewer #2: Dear authors, your study is fairly conducted and the results are well presented. Still, I have some concerns regarding the fact that you did not evaluate the type of thyroid cancer, given the fact these types have many differences in growing behavior. Maybe you can address this in a paragraph, searching if there are certain types of thyroid cancer that are more prone to develop. Also, I think you have to write a paragraph about the strengths of this study; you have already stated the limitations, so a section on strengths should be welcomed.
Good luck!
We want to thank the reviewer for providing a comprehensive review of our manuscript and recommendations to improve our study. Below, you will find our point-by-point answers to the comments raised by the reviewer. The provided page numbering refers to the clean version of the revised manuscript.
- I have some concerns regarding the fact that you did not evaluate the type of thyroid cancer, given the fact these types have many differences in growing behavior. Maybe you can address this in a paragraph, searching if there are certain types of thyroid cancer that are more prone to develop.
Thank you for this thoughtful comment. We agree that the study would be stronger if histology type would be included. Unfortunately, CDC only include all thyroid cancer histology and no distinction of different types of thyroid cancer is made. We believe that it would be beneficial to examine lindane exposures and its association with different aggressive variants. We have addressed this as a limitation to our study in the discussion section (page 8) In addition, we did a thorough search to see if certain types of thyroid cancer are more prone to develop after lindane exposure, but unfortunately there is no information on this.
“. In addition, CDC only include all thyroid cancer histology types and no distinction of different types of thyroid cancer is made. It would be beneficial for future studies to examine lindane exposure and its association with different aggressive histological variants.”
- Also, I think you have to write a paragraph about the strengths of this study; you have already stated the limitations, so a section on strengths should be welcomed.
We agree with the reviewer’s point that strengths of this study should be included. We have added a paragraph addressing the strengths to the discussion section (page 8).
“To the best of our knowledge, this is the first ecological study investigating the association between lindane exposure and thyroid cancer risk in large representative populations. Therefore, we believe our report provides valuable new information in relation to the effects of lindane use. Strengths include the large number of states included in the study, despite some missing data for a few states”.
Reviewer 3 Report
The article entitled “Correlation between lindane use and the incidence of thyroid cancer in the United States; an ecological study” has been evaluated. The aim of this ecological study was to assess the potential correlations between lindane exposure and thyroid cancer incidence in the United States (US). Data on statewide age-adjusted thyroid cancer incidence rate (per 100,000 people) was obtained from the Centers for Disease Control and Prevention for all US states for 2019. Lindane use estimates per cropland (kg/acres) were then overlaid on the map of age-adjusted thyroid cancer incidence rate using ArcGIS. The trend of lindane use in the US between 1992 and 2007 was calculated using the Mann-Kendall correlation test. The correlation between statewide lindane use and age-adjusted thyroid cancer incidence rates was calculated using Spearman correlation.
Major Revision:
1. The authors mentioned that they performed the Mann-Kendall correlation test. It should be appropriate to include a Table for this calculation.
2. Author made a conclusion based on the Spearman correlation. It would be a good idea to perform any alternative correlation to validate this result.
Author Response
Reviewer#3: The article entitled “Correlation between lindane use and the incidence of thyroid cancer in the United States; an ecological study” has been evaluated. The aim of this ecological study was to assess the potential correlations between lindane exposure and thyroid cancer incidence in the United States (US). Data on statewide age-adjusted thyroid cancer incidence rate (per 100,000 people) was obtained from the Centers for Disease Control and Prevention for all US states for 2019. Lindane use estimates per cropland (kg/acres) were then overlaid on the map of age-adjusted thyroid cancer incidence rate using ArcGIS. The trend of lindane use in the US between 1992 and 2007 was calculated using the Mann-Kendall correlation test. The correlation between statewide lindane use and age-adjusted thyroid cancer incidence rates was calculated using Spearman correlation.
We want to thank the reviewer for providing a comprehensive review of our manuscript and recommendations to improve our study. Below, you will find our point-by-point answers to the comments raised by the reviewer. The provided page numbering refers to the clean version of the revised manuscript.
- The authors mentioned that they performed the Mann-Kendall correlation test. It should be appropriate to include a Table for this calculation.
|
Year |
Mann-Kendall |
p-value |
|
1992-2007 |
T= -0.617 |
<0.001 |
We agree with the reviewer that a table should be included with the results from the Mann-Kendall analysis. We have included the table as table 2.
- Author made a conclusion based on the Spearman correlation. It would be a good idea to perform any alternative correlation to validate this result.
We thank the reviewer for this suggestion. The ecological nature of our paper validates the use of the spearman correlation. We will explore other databases to investigate the association between Lindane exposure and thyroid cancer for a future research paper. We appreciate this suggestion.
Round 2
Reviewer 3 Report
The authors did significant changes as per the reviewer's suggestion. The MS can be acceptable for publication.